# The Role of Nutraceuticals and Phytonutrients in Chickens’ Gastrointestinal Diseases

**DOI:** 10.3390/ani12070892

**Published:** 2022-03-31

**Authors:** Lucia Biagini, Livio Galosi, Alessandra Roncarati, Anna-Rita Attili, Sara Mangiaterra, Giacomo Rossi

**Affiliations:** School of Biosciences and Veterinary Medicine, University of Camerino, 62024 Matelica, Italy; alessandra.roncarati@unicam.it (A.R.); annarita.attili@unicam.it (A.-R.A.); sara.mangiaterra@unicam.it (S.M.); giacomo.rossi@unicam.it (G.R.)

**Keywords:** poultry, antibiotic alternatives, nutraceuticals, phytonutrients

## Abstract

**Simple Summary:**

The use of nutraceuticals and phytonutrients in poultry nutrition has been extensively explored over the past decade. The interest in these substances is linked to the search for natural compounds that can be effectively used to prevent and treat some of the main diseases of the chicken. The serious problem of antibiotic resistance and the consequent legislative constraints on their use required the search for alternatives. The purpose of this review is to describe the current status of the effects of some substances, such as probiotics and prebiotics, organic acids, vitamins and phytogenic feed additives, focusing specifically on studies concerning the prevention and treatment of four main gastrointestinal diseases in chicken: salmonellosis, necrotic enteritis (caused by *Clostridium perfringens*), campylobacteriosis, and coccidiosis. A brief description of these diseases and the effects of the main bioactive principles of the nutraceutical or phytonutrient groups will be provided. Although there are conflicting results, some works show very promising effects, with a reduction in the bacterial or protozoan load following treatment. Further studies are needed to verify the real effectiveness of these compounds and make them applicable in the field.

**Abstract:**

In poultry, severe gastrointestinal diseases are caused by bacteria and coccidia, with important economic losses in the poultry industry and requirement of treatments which, for years, were based on the use of antibiotics and chemotherapies. Furthermore, *Salmonella* spp., *Clostridium perfringens*, and *Campylobacter jejuni* can cause serious foodborne diseases in people, resulting from consumption of poultry meat, eggs, and derived products. With the spread of antibiotic resistance, which affects both animals and humans, the restriction of antibiotic use in livestock production and the identification of a list of “critically important antimicrobials” became necessary. For this reason, researchers focused on natural compounds and effective alternatives to prevent gastrointestinal disease in poultry. This review summarizes the results of several studies published in the last decade, describing the use of different nutraceutical or phytonutrients in poultry industry. The results of the use of these products are not always encouraging. While some of the alternatives have proven to be very promising, further studies will be needed to verify the efficacy and practical applicability of other compounds.

## 1. Introduction

For more than 60 years, antibiotics have been used in livestock without any specific control. In the poultry industry, these have been employed for different purposes: treatment of various pathologies (therapy), their prevention (metaphylaxis), and especially as growth promoters [1]. The frequent use of subtherapeutic antibiotic doses, administered to the whole flock in large quantities, has made a great contribution to the development of antibiotic resistance. According to the World Health Organization, it represents “one of the greatest threats to global health, food security and development” today [2], as the connection between the use of sub-therapeutic doses and the development of resistance between different classes of antibiotics has been proven [3]. Even more important, there is clear evidence of the adverse consequences for human health caused by resistant organisms deriving from non-human usage of antibiotics, primarily *Salmonella* spp. [4]. For this reason, in 2006 the European Union definitively banned the use of antibiotics as growth promoters in animal feed, describing it as “the final step in the phasing out of antibiotics used for non-medicinal purposes” [5,6]. Subsequently, this ban was also approved in the United States in 2017 [7]. However, the use of antibiotics is still allowed for therapeutic purposes, and the appropriate and reasonable use is a duty of both human and veterinary medicine. Guidelines for the choosing of therapeutic action were written by the World Health Organization, highlighting which are the critically important antimicrobials and why [4].

Poultry meat production is always growing, with a new high in the European production of 13.6 million tonnes for the year 2020 [8], giving even more importance to the research of alternatives to the use of antibiotics. This review will focus on the four most important and serious gastrointestinal (GI) diseases of the chicken: *salmonellosis*, *clostridiosis* (Necrotic enteritis), *campylobacteriosis*, and *coccidiosis*. In all these pathologies, there is an increasing resistance to different classes of antibiotics. Even more important, except to coccidiosis, some species of *Salmonella* spp., *Clostridium* spp., and *Campylobacter* spp. are causes of severe human foodborne diseases. For these reasons, the reduction in circulation of these pathogens in farms and the consequent contamination of meat and eggs it is of fundamental importance. In this paper, the main natural alternatives to antibiotics included in the nutraceutical and phytonutrients groups are reviewed. The potential of these substances to directly reduce the pathogenic load or limit the effects of GI infection in chicken will be described, highlighting the underlying mechanisms of action.

## 2. Different Classes of Alternative Compounds

Gut health is essential in the pathogenesis of different intestinal disorders. The role of gut microbiota has been well described, showing how dysbiosis can be an important predisposing factor for GI diseases [9]. Different factors are able to influence gut microbiota such as antibiotics, diet, or pathogenic infection. A perfect alternative to antibiotics should be able to prevent various pathogens’ infection or reduce their effect, considering clinical manifestation and organ damages. Ideally, an alternative should have the same mechanism proposed for antibiotic growth promoters in terms of microbiota modulation and immunomodulation [10], while also having a growth-promoting effect. Indeed, growth promotion was one of the main reasons that led to the use of subtherapeutic doses of antibiotics in chicken farming. In the last decade significant work has been contributed for the research of alternatives to antimicrobials, especially referring to substances included in the groups of nutraceuticals and phytonutrients. The term “nutraceutical” was created in 1989 by Stephen DeFelice, combining the terms “nutrition” and “pharmaceutical” [11]. According to DeFelice, nutraceuticals can be defined as “a food (or part of a food) that provides medical or health benefits, including the prevention and/or treatment of a disease” [12]. Different products, all natural, are utilized as nutraceuticals: dietary fibre, probiotics, prebiotics, organic acids, antioxidants, vitamins, polyphenols, and spices [13]. With the term “phytochemical” instead are described “metabolites from plants, including mostly plant secondary metabolites” [14]. In this term are also enclosed the terms phytonutrient and bioactive, and it can be considered part of nutraceutical group. There is still a big debate on the most appropriate use of these terms and nomenclature, and especially for plant-derived food components there is a lack of standardization in definitions (Table 1) [14]. The purpose of their use in the poultry sector is linked to the possibility of obtaining a regulation of the composition of the intestinal microbiota, improving GI health, the functionality of the intestinal barrier, and the activity of the host’s immune system with an effect also on weight gain and feed conversion ratio [15,16]. Probiotics, prebiotics, organic acids, vitamins, enzyme, phytobiotics, and phytochemicals are included in this group (Table 2). All of these compounds are usually administered by feed, water, or in ovo.

## 3. *Salmonella* spp. Infection

The genus *Salmonella* is part of the family *Enterobacteriaceae* and comprises three species: *S. enterica*, *S. bongori*, and *S. subterranea*. The species *S. enterica* includes six subspecies, but only one (*S*. *enterica* subspecies *enterica*) is associated with the development of disease in warm-blooded animals. In the subspecies *S. enterica*, the serovars Enteritidis and Typhimurium are most prevalent both in humans and in poultry [55,56]. *Salmonella enterica* serovar Enteritidis is commonly associated with poultry and derived products, whereas serovar Typhimurium has a wider species range, affecting pigs and cattle as well as poultry [57]. This bacterium has a worldwide distribution and causes big losses in poultry industries. Generally, clinical signs of *Salmonella* infection are evident only in young chickens that show depression, drooping wings, ruffled feathers, anorexia, emaciation, and watery diarrhoea. The peak of morbidity and mortality is usually around the first 2–3 weeks of life when weight loss or growth retardation are observed, while clinical signs are rare in older birds. [55]. It also has a big relevance for human health considering that, in Europe, *S. enterica* is the second most common foodborne disease [58]. Multiple epidemiological studies reveal the role of poultry meat and eggs in outbreaks of human salmonellosis [3,59,60,61].

Over the years, antibiotics have been extensively used to treat this pathology, and their improper use has favoured the development of multidrug-resistant strains. The emergence of resistance started at first with antibiotics of older use (ampicillin, chloramphenicol, trimethoprim-sulfamethoxazole) and subsequently involved fluoroquinolones (ciprofloxacin) and extended-spectrum cephalosporins [57]. The problem of resistance concern “critically important antibiotics for human health”, leading to the request of new molecules for therapy (e.g., carbapenems) [57]. Consequently, the spread of *Salmonella* infection cannot only be treated using antibiotics, as this could promote the development of pathogenic strains as well as led to the presence of antibiotic residues in poultry meat [62]. In order to limit this use as much as possible, there is continued research on sustainable and safe alternatives to administer in poultry.

### 3.1. Probiotics and Prebiotics

A considerable amount of studies describe the use of different probiotic strains to increase the resistance against salmonellosis in poultry [63,64]. The idea of using probiotics to obtain a microbial control is linked to the concept that a healthy microflora can inhibit pathogens’ colonization through a mechanism of competitive exclusion, where the probiotic can compete for intestinal space, reducing the chance of pathogen colonization [65]. This concept is valid for *Salmonella* spp. as far as for other pathogens that will be discussed later. However, other mechanisms are also involved in probiotic effects such as reduction in intraluminal pH due to production of short chain fatty acids, production of antimicrobial peptides, optimization of intestinal functionality and activation of immune response. Researchers tested single or multi-strain probiotics, in particular *Lactobacillus*, *Enterococcus*, and *Bacillus* strains, with encouraging results after a *Salmonella* challenge in chickens, considering parameters such as bacterial count in ceca, faecal bacterial load, growth performance, and immune functionality [56,66,67,68,69,70,71]. The effect of commercially available probiotic supplements such as EarlyBird^®^ (Pacific Vet Group USA Inc., Fayetteville, AR, USA) and FloraMax-B11^®^ (Pacific Vet Group USA Inc., Fayetteville, AR, USA) alone, or with the addition of glycerol, was tested to obtain a protective effect against *Salmonella* Enteritidis colonization, demonstrating the prevention of intestinal colonization from the pathogen [72,73]. In contrast, Khan and Chousalkar show that the administration of probiotics is not able to reduce shedding and invasion of *Salmonella* spp. in chickens [74]. The variability of the results is understandable as the works differ for bacteria strains, concentration, period, and duration of administration.

Studies on prebiotics mainly focus on the use of oligosaccharides such as mannanoligosaccharides (MOS), galactooligosaccharides (GOS), fructooligosaccharides (FOS), and inulin. The use of MOS can inhibit the activity of *Salmonella* spp., reducing the adhesion to intestinal epithelium thanks to the presence of mannose in the lumen [34] and increase the immunity response against *S.* Enteritidis with higher T lymphocyte infiltration in intestinal mucosa [75]. The commercial product XPC^®^ (Diamond V, Cedar Rapids, IA, USA) reduced faecal *Salmonella* spp. count, with an increase in butyrate concentration in GI tract [76,77]. Lee et al. [78] evaluated the effect of the commercial prebiotic Biolex^®^ MB40 (Andersen, Barcelona, Spain), registering a not significant reduction in *Salmonella* spp. counting. Similar inconclusive results are obtained with the use of a prebiotic GOS, however an increase in the gene expression of the cecal tonsils and an influence in the composition of the cecal microbiome were recorded, suggesting the usefulness of this treatment [79].

### 3.2. Organic Acids

Organic acids have gained attention as a possible alternative to antibiotics. Previous studies showed their multiple effect on GI tract such as increase in growth performances, improved nutrient metabolism, anti-inflammation effects, and reduction in *Salmonella* spp. colonization [80,81]. Treatments with organic acids, whether they are SCFA, MCFA, or other organic acids, have a different powerful antimicrobial activity, depending on whether they are used individually or in mixtures, and on their concentration [52]. In particular, better result on decreasing *Salmonella* spp. colonization can be obtained using coated acids [82]. Focusing on propionic and fumaric acids, crop and gizzard are the site in which the greater concentration is obtained after oral administration. Although the crop is an initial site for the settlement of infection, the sites in which there is a greater colonization of *Salmonella* spp. are the ceca, and it is important that organic acids can reach the low intestinal tracts in order to have a better effect on animal health. Uncoated butyrate has a faster absorption and is therefore not able to reach this site [82]. For this reason, recent studies on the use of butyrate are especially focused on the coating and inclusion technique [83,84]. Feeding broiler with butyrate included in a wax matrix significantly reduces *Salmonella* spp. colonization in ceca content [81]. Not only butyrate, but also other organic acids, have been added in feed or water. An organic acid blend composed of formic acid and sodium formate mixture (Amasil^®^ NA, BASF, Ludwigshafen, Germany) permits us to obtain a significant effect on reducing *S.* Thyphymurium infection in broilers [85]. A decrease in *Salmonella* spp. cecal count can also be achieved by using a feed additive mixture containing organic acids and ß-1,4 mannobiose [86]. Furthermore, it has been shown that adding formic acid to broiler feed appears to prevent *Salmonella* spp. passing from challenged to non-challenge sheds, without, however, having a reduction in *Salmonella* spp. counting [87]. Dietary supplementation with a symbiotic and an organic acid can also be used to improve growth performance and reduce carcass *Salmonella* spp. in broilers [88].

### 3.3. Vitamins

Another possible way to control *Salmonella* spp. infection in broilers is the use of vitamins, C and E in particular. Vitamin C can alleviate the effects of multiple stressors in animals and, alone or with other compounds such as curcumin, allows a reduction in *Salmonella* spp. count [89,90,91] and an improvement of the intestinal health [92]. Feed supplementation with vitamin E results in reduction in oxidative and immune stress that occurs during the infection [93] and its combination with arginine increases resistance against bacterial colonization, although there is no reduction in the concentration of *S.* Typhimurium in ceca [94].

### 3.4. Phytogenic Feed Additives (PFAs)

PFAs have gained interest due to their ability to help maintaining a healthy gut environment. It has been reported that essential oils of herbs and spices can play a significant role in bird health and performance by stimulating feed intake, secretion of endogenous enzymes, production of antioxidants, and antibacterial effect [95]. Included in this group are plant extracts and their active ingredients, whose beneficial qualities are linked to some bioactive molecules contained in it such as carvacrol, thymol, capsaicin, cineole, etc. [96]. The comparative effects of antibiotics and different PFAs, such as thymol essential oil, thyme essential oil, anise, and other components, on *Salmonella* Typhimurium-challenged broilers shows promising results [95]. A PFA containing extract of fennel (*Foeniculum vulgarae* var. *dulce*), lemon balm (*Melissa officinalis*), peppermint (*Mentha arvensis*), anise (*Pimpinella ani-sum*), oak (*Quercus cortex*), cloves (*Syzygium aromaticum*), and thyme (*Thymus vulgaris*) was tested in broilers challenged with *Salmonella* spp., proving that it can be considered as an alternative to improve the growth performances of broilers when exposed to infection [97]. Additionally, the effect of chestnut and quebracho wood was evaluated, showing a reduction in both mortality and *Salmonella* spp. excretion [98].

## 4. *Campylobacter jejuni* Infection

Thermophilic *Campylobacter* spp., primarily *Campylobacter jejuni* and *C. coli*, are colonizers of the intestinal tract of domestic poultry such as chickens and turkeys. These bacteria have a worldwide distribution in poultry flocks, causing little or no clinical symptoms [99] although clinical forms with watery, mucoid, or bloody diarrhoea, damage and inflammation of the mucous membrane, weight loss, and mortality were demonstrated in challenged young chickens [100]. There is a natural faecal–oral transmission of *Campylobacter* spp., which establishes in the intestinal tract with higher load in caeca [101]. However, it represents a main issue for human health; in fact, in the European Union, it represents the first zoonoses reported in a human in 2020 [58]. For this reason, although *Campylobacter* spp. is not a main issue for poultry, its control is important for food safety and human health. In human, *C. coli* and, mainly, *C. jejuni*, are cause of foodborne gastroenteritis with symptoms such as watery or bloody diarrhoea, fever, abdominal cramps, and possible severe condition in immunocompromised patients and with correlation with *Guillain–Barré* syndrome (a paralytic autoimmune complication) [99]. These infections are due to carcass contamination during slaughtering and spread to poultry meat and subsequently to consumers [102]. A correlation between the degree of intestine invasion and the level of contamination of the carcass has been demonstrated [103]. Other important elements in the attempt to reduce the spread of *Campylobacter* spp. are identifying possible sources of contamination and avoiding the persistence in the environment caused by contaminated litter, rodents, flies, other animals, short interruptions in production, inadequate disinfection, and contamination of the water and surrounding environments [104]. Against this pathogen, the intervention strategies are focused on prevention of colonization and/or its reduction. Nutraceuticals can have a great potential in order to avoid the use of antibiotic.

### 4.1. Probiotics and Prebiotics

The use of probiotics and prebiotics have been investigated to prevent *Campylobacter* spp. colonization of the GI tract. The genera of probiotic most commonly evaluated against *C. jejuni* are *Lactobacillus* spp., *Bacillus* spp., and *Enterococcus* spp. [105]. The interest in *Lactobacillus* spp. is based on the ability to reduce intraluminal pH of GI tract, creating an inhospitable environment for other bacterial species [106,107,108]. The ability of probiotic strains to reduce *C. jejuni* count is due to a reduction in its adhesion ability to epithelial cells, that prevent the colonization [109,110,111,112,113]. A significant reduction in caecal *Campylobacter* spp. count is registered after the administration of *Butyricicoccus pullicaecorum*, a probiotic able to produce butyrate [114]. On the contrary, it has been found that a competitive exclusion mixture could not compete against *C. jejuni* in challenged broilers [115]. Smialek et al. [105] suggest that a multispecies preparation may have a higher activity against *Campylobacter* spp. than single one strain probiotic. Few studies also refer to the use of prebiotics, although they are usually in association with probiotics, because they probably cannot be efficacious on their own. A symbiotic product formulated with microencapsulated probiotic *Bifidobacterium longum* PCB133 and a xylo-oligosaccharide (XOS) showed to be more effective in reducing *C. jejuni* load in ceca when the product is given over lifelong treatment in comparison to shorter administration [116]. Other similar studies support the use of symbiotics against *C. jejuni,* also investigating the efficacy of inhibition obtained using *Bifidobacterium* spp. and *Saccharomyces cerevisiae* [22,111,117]. An increased antimicrobial activity against *Campylobacter* spp. is observed using a combination of *L. casei* and berry pomace phenolic extract (BPPE), because bioactive phenols can stimulate the activity of the probiotic bacteria and its metabolites while inhibiting pathogens growth [118].

Probiotics can also be applied in combination with vaccines used against *Campylobacter* spp. infection. The use of the probiotic *Anaerosporobacter mobilis* or *L. reuteri* in broilers and Leghorn layer chickens is able to increase the immune response to the vaccination [119].

### 4.2. Organic Acids

Another strategy for the control of *C. jejuni* spread is the use of organic acid during the broiler rearing phase. The mechanism of action of organic acid is not currently well described, but it is known that, after administration, organic acids enter the cytoplasm, altering the equilibrium of cellular hydrogen that causes an inhibition of essential metabolic cellular reactions and accumulation of toxic anions [120].

Some studies have been conducted to assess the reduction in *C. jejuni* colonization on chickens during breeding and preslaughter phases, with positive results after experimental challenge [110,121,122]. Recently, Peh et al. showed in vitro synergistic activities of a combination of caprylic, sorbic, and caproic acid against the major *Campylobacter* species, which could also be promising for an in vivo approach [123]. MCFA added to feed or water can limit *C. jejuni* colonization [124,125]. However, there are still conflicting results regarding the effects of this administration, because a good efficacy in vitro is not followed by the same effects during in vivo trials [126,127].

### 4.3. Phytogenic Feed Additives (PFAs)

The use of phytogenic feed additives (PFAs), as essential oils, tannins, and plant extract, is well explored. For compounds of plant origin, most of the studies are initially carried out in vitro, with good results which, however, are often not found after in vivo administration. For example, for cinnamon oil ingredient trans-cinnamaldehyde (CIN) and allicin, a compound extracted from garlic, the efficiency observed in vitro did not allow a reduction in colonization in vivo [128,129]. An in vitro antimicrobial activity has also been described using an extract containing hydrolysable and condensed tannin [130]. Several studies describe the use of essential oils, plant extracts, and secondary plant compounds, without a marked effectiveness against *C. jejuni* [131,132]. Some good results are described with the use of 0.25% thymol, 2% thymol, 1% carvacrol, and 0.5% thymol and carvacrol [133]. Different essential oils, polyphenol, and terpenoid compounds were tested against *C. jejuni*, and a strong activity of essential oils and terpenoid compound was reported [134].

## 5. *Clostridium perfringens* Infection

Necrotic enteritis (NE) is a disease caused by the toxins produced by pathogenic strains of *Clostridium perfringens* type A, C, and G, that represent a major cause of losses in the poultry industry. Symptoms of NE are very non-specific, such as depression, diarrhoea, ruffled feathers, anorexia, and dehydration. In acute forms, animals can die without clinical signs. More frequently, subclinical forms of NE causes only a reduction in feed intake and weight. Macroscopic lesions observed during autopsy are characteristic, with the small intestine becoming fragile, hyperaemic, and dilated for the presence of gas. On the mucosal surfaces, light brown pseudo-membranes and occasional bleeding are present [135,136]. *C. perfringens* is normally found in the GI content of healthy chickens [137], and the development of clinical forms is linked to predisposing factors such as coccidiosis, reduction in feed quality, or presence of other immunosuppressive disease [135]. *C. perfringens* is also a public health issue due to its ability to produce enterotoxin at the moment of sporulation, causing a foodborne illness in humans, with subtype A that gives diarrhoea and subtype C that causes NE [135]. In the past years, a correlation between outbreaks of subtype A illness in humans and chicken meat consumption has been demonstrated [138]. Traditional strategies to control NE rely on prevention and direct treatment in case of clinical form. Antibiotics, and especially lincomycin, bacitracin, and tylosin, have been widely used. However, the first fundamental element to limit the spread and clinical manifestations of NE lies in obtaining the maximum reduction in predisposing factors, primarily coccidiosis. For this reason, new forms of action against this pathogen are necessary.

### 5.1. Probiotics and Prebiotics

The importance of the use of probiotics, prebiotics, and symbiotics is aimed not only at achieving a reduction in *C. perfringens* infection, but also at improving intestinal health, limiting the dysfunctions due to lesions of intestinal tight junction, and consequently alteration of nutrient absorption and/or bacterial translocation. The use of *Bacillus subtilis* PB6 significantly ameliorates intestinal morphology, increasing villus length and villus length/crypt depth ratio in infected chickens [139]. Multiple strains of *Bacillus* spp. showed an agonistic activity against *C. perfringes* thanks to the production of bacteriocins and other antimicrobial peptides, with significant attenuation of *C. perfringes* symptoms [140,141,142,143,144]. The commercial product FloraMax-B11^®^ (Vetanco, Villa Martelli, Argentina) tested on chickens challenged with *E. maxima*, *S. typhimurium*, and *C. perfringens*, shows a reduction in intestinal lesion and *C. perfringens* count [145]. Different *Lactobacillus* strains reduce *C. perfringens* pathological effects [146,147,148]. The yeast extract NuPro^®^ (Alltech, Nicholasville, KY, USA) administered on chicks challenged with *C. perfringens* shows a reduction in intestinal lesion score [130]. Similar results are obtained using other commercial yeast additives such as Safmannan^®^ (Phileo by Lesaffre, Marcq-en-Barœul, France) [149] or the yeast cell wall extract from *Saccharomyces cerevisiae* (Actigen^®^, Alltech, Nicholasville, KY, USA) [150], with promising results.

### 5.2. Organic Acids

SCFA (formic, acetic, propionic, and butyric) and MCFA (caproic, caprylic, and capric acids) are described as promising alternatives to antibiotics in *C. perfringes* infections. These additives provide evidence of reducing the negative effects of pathogen proliferation, such as reduction in weight gain and GI disfunction [151]. Some mixtures of organic acids have been tested, alone or blended with essential oils, showing a potential antimicrobial and protective activity, with a reduction in intestinal lesions [152,153,154,155]. There are still several factors influencing the effects of these products such as their structure, coating, dosage, dietary composition, and environmental condition [153].

### 5.3. Phytogenic Feed Additives (PFAs)

The study of PFAs is particularly developed with focus on the effect on *C. perfringens* intestinal burden and intestinal gross lesion. Essential oils (EOs) have a major component in phenolic compounds such as thymol, carvacrol, and eugenol, that showed to have a strong antibacterial activity [153]. The use of essential extracts from *Origanum vulgare*, *Piper nigrum*, *Syzygium aromaticum*, and *Thymus vulgaris*, and their components (thymol, carvacrol, and eugenol) against *C. perfringens* has been explored and, although the mechanism of action of this substances is not still well understood, in some cases a direct inhibitory effect on the pathogen or an action against its toxins is described [156]. The effects of thymol and carvacrol essential oils and lysozyme were tested, suggesting that both have positive effects, but that their blend does not improve their effects [157]. Additionally, tannins, and in particular two common sources of tannins, chestnut (*Castanea sativa*) and quebracho (*Schinopsis lorentzii*) extracts, have an activity against *C. perfringens* and its toxins, reducing the severity of intestinal damage and bacterial count and protecting infected intestinal tissues from oxidative damage [158,159,160].

### 5.4. Vitamins

Vitamins can have a preventive activity in chicken with NE, despite this field being not well explored. In broilers, lesion score due to *C. perfringens* infection and *C. perfringens* count in intestine were reduced after treatment with beta-carotene [161].

## 6. Coccidiosis

Coccidiosis is a poultry disease of universal importance. It is caused by parasites of the genus *Eimeria*. Poultry are susceptible to seven species of *Eimeria* (*E. acervuline*, *E. maxima*, *E. brunetti*, *E. praecox*, *E. mitis*, *E. tenella*, and *E. necatrix*), with most serious condition described consequently to *E. tenella* and *E. necatrix* infections [162]. *Eimeria* spp. causes intestinal damage with impaired digestive process, loss of nutrients absorption capability, dehydration, and increased susceptibility to other pathological agents. Cases of severe outcomes are associated with bloody diarrhoea and very high mortality [162]. With the use of anticoccidial drugs, mainly sulphonamides, there has been a reduction in severe clinical manifestations and now *Eimeria* spp. infections are mostly associated with subclinical manifestations. However, it still remains the cause of important economic losses for poultry industries. Moreover, *Eimeria* spp. causes changes in permeability and functionality of the intestinal mucosa, being considered one of the main predisposing factors for bacterial infections [163]. It is a ubiquitous and resistant parasite, so prevention is the most important strategy in poultry farming. Furthermore, as for antibiotics, anticoccidial drugs also saw the development of tolerance with widespread resistance. Vaccination against several *Eimeria* spp. was very promising, but side effects such as post-vaccination mild infection and reduction in weight gain and feed conversion are discouraging this practice, and the interest towards different nutraceutical and phytochemical remedies with anticoccidial properties is increasing [162].

### 6.1. Probiotics

Over the past years, many compounds containing one or more bacterial strains such as *Bacillus* spp., *Lactobacillus* spp., *Enterococcus* spp., *Pedicoccus* spp., and *Bifidobacterium* were tested with very promising results. Probiotic bacteria prevent Eimeria invasion by adhering to the intestinal mucosa, thus reducing receptor availability during Eimeria spp. infection. This limits the perforation and secretion of sporozoites in the intestinal mucosa, allowing a reduced proliferation and spread of the oocysts [164]. Moreover, probiotic bacteria have an immunomodulating and antioxidant effect, increasing GI microbiota balance and improving intestinal functionality and health [165]. Some commercial probiotics mixture such as PoultryStar^®^ (DSM, Heerlen, The Netherlands) [166,167], Smart ProLive^®^ (Bakın Tarım, Ankara, Turkey) [168], and Primalac^®^ (Star-Labs, Clarksdale, MO, USA) [169], have been able to guarantee a reduction in the intestinal lesion score, higher growth rate, and reduced oocyst shedding.

Probiotics can be administered together with the vaccine in the attempt to eliminate its side effects. Despite vaccines being considered relatively effective for the control of this disease, probiotic addition could indeed enhance animals performance and give a strong protective effect in *Eimeria* spp. challenged chickens with an improvement of the immune response [167,170]. There are still conflicting results on the effectiveness of administration in reducing infection and symptoms of coccidiosis. A recent study compares the effect of probiotic, prebiotic, salinomycin, and vaccine, suggesting that probiotics and prebiotics are not as effective in controlling coccidiosis and its complications as vaccine or salinomycin [171].

Additionally, some prebiotics have been tested for poultry coccidiosis treatment such as inulin, fructo-oligosaccharides, mannan-oligosaccharides (MOS), and xylo-oligosaccharides [85]. The commercial prebiotic Fermacto^®^ (Pet-Ag, Hampshire, IL, USA), derived from *Aspergillus orizae*, was evaluated in different *Eimeria* spp. infections, with very promising results [171]. Furthermore, it has been shown that both the administration of a prebiotic (mannan-oligosaccharides and β-glucans) and a *Bacillus subtilis* probiotic do not cause negative interactions with the vaccination for coccidiosis and indeed are able to increase the feed conversion ratio [172]. Similarly, the comparison of the effects of MOS and Amprolium (an anticoccidial chemoterapic) administration on performance and GI health of broiler challenged with *E. tenella*, suggests that MOS are able to improve growth performance and reverse *E. tenella* lesions [173].

### 6.2. Organic Acids

The preventive anticoccidial activity of organic acids was mainly attributed to their ability to lower ceca pH and induce protective immunity against *Eimeria* spp. [174]. Anticoccidial properties of acetic acid against *E. tenella* were described [85], and its effects compared with Amprolium show almost equivalent results in reducing negative consequence of infection, demonstrating the potential of acetic acid use as an alternative to chemotherapy [175]. Interesting results are also reported for the use of glycerol monolaurate, obtained from lauric acid and glycerol, [176] and for butyrate, clopidol, and their combination [177], with a reduction in coccidian infection and maintenance of the immunity obtained from the first infection. The use of a blend of benzoic acid and essential oil compound is reported in animals challenged with *Eimeria* spp., resulting in a reduction in intestinal lesions [178].

### 6.3. Phytogenic Feed Additives (PFAs)

PFAs, together with probiotics, are the most interesting natural alternatives in coccidiosis treatment. Most of the plants and their bioactive compound used against *Eimeria* spp. infection have been reviewed by El-shall et al. [179]. From this comprehensive review clearly emerges the big interest and potential of phytochemicals in poultry industry, with a huge amount of papers on this topic. In most cases, not only herbs, but also herbal mixtures have been shown to be effective against avian coccidiosis [179]. The mechanism of action is not always known, but some anticoccidial effects were identified: inhibition of different *Eimeria* spp. growth, prevention of invasion, strengthening of immune response, inhibition of sporulation, prevention of oocysts shedding, and reduction in oocyst score. The herbal extracts and their phenolic compounds react with *Eimeria* cell membrane causing cell death. Moreover, these extracts increase the intestinal lipid peroxidation, enhance the reparation of injured epithelium, and decrease the permeability of intestinal cells induced by *Eimeria* spp. with a higher cellular turnover [179].

### 6.4. Antioxidants

The addition of vitamins in broilers’ diet is a strategy described to limit the consequences of *Eimeria* spp. infection as it can cause a reduction in cellular content of some antioxidant vitamins in the host cells [174]. Moreover, the limitation of peroxidation is the way of action of some anticoccidial molecules such as toltrazuril and salinomycin [174]. Supplementation with selenium, zinc, vitamin E, copper, and manganese can mitigate the effect of the disease [143,180,181], and the use of a blend of curcumin (*Curcuma longa*) and microencapsulated phytogenic, containing thymol, cinnamaldehyde, and carvacrol, permits a decrease in coccidian load [182]. Furthermore, feeding vitamin E and arginine to poultry, at higher level than recommended, can improve innate and humoral response against *Eimeria* spp. challenged animals [183]. Additionally, curcumin and cinnamaldehyde can protect intestinal cells from lipid peroxidation caused by coccidian parasites, increasing antioxidant enzyme levels [184].

## 7. In Ovo Technique

The in ovo inoculation is a method which allows us to administer substances directly to the chicken embryo during the incubation period. It was originally designed for the Marek’s Disease (MD) vaccine to achieve an early and effective immunity [185]. This technique spread rapidly worldwide, especially after the invention of the first automated injection system, the Inovoject^®^ machine, manufactured by Embrex, Inc. (Durham, NC, USA) and initially introduced in the North American poultry industry [186]. The in ovo technique was suddenly explored for the administration of many other different compounds. The concept of in ovo feeding was introduced in 2003 as the administration into the embryonic amnion of nutrients and other natural compounds that can modulate enteric development of the hatchling [187]. Chickens hatch with an incomplete development of the intestinal tract and, consequently, the period from hatching to the first feeding is very critical, so the administration of substances in ovo allows us to obtain important effects on the future animal growth. Injections can be performed in different sites (air chamber, allantoid sac, amniotic sac) and day of incubation (from 12 to 19 day). The preferred site for inoculation is the amniotic sac, where the MD vaccine is also administered, and the inoculated substances are ingested by the chicken before hatching, coming into direct contact with the digestive and respiratory systems [188]. Over the years, this method has been used to deliver various nutrients, vaccines and drugs and it has also been explored as a possible way to obtain an early protection against pathogens, through multiple mechanism (Figure 1), and especially allowing a correct microbiota colonization of the GI tract [189,190] enhancing the development of both GI and immune system [23], allowing an early interaction with the chickens immune system before hatching [191].

### Using *in Ovo* Inoculation Technique against GI Pathogens

The contact with environmental *Salmonella* spp. often occurs before the chicken has consumed its first feed. Probiotic bacteria can prevent pathogens’ colonization, with the mechanism of competitive exclusion [192] considering also that a well-developed GI tract can reduce the effect of a mild *Salmonella* spp. colonization [190]. In this context, in ovo administration of probiotic and prebiotic is able to reduce *Salmonella* colonization and faecal shedding, with the amniotic sac as preferred site of administration, as the inoculum goes in direct contact with the GI system [193]. The in ovo treatment with *Enterococcus faecium* resulted in significant effects: after a challenge carried out at 4 days of age, a reduction in the number of chickens positive for *S.*
*enteritidis* was observed, with an increased effect continuing the administration of the probiotic in diet [190]. Similar results are described after the in ovo administration of the commercial probiotic mixture FloraMax^®^-B11 (Pacific Vet Group USA Inc., Fayetteville, AR, USA), revealing an increase body weight, higher villi surface area, and decreased *S.* Enteritidis recovery after challenge at 7 day of age [194]. Administration of probiotics in ovo can also be effective to decrease *Salmonella* colonization in ceca [195,196].

The use of nutraceuticals and phytonutrients in ovo to prevent *Campylobacter jejuni* infection is still not explored, but some trials have been performed in relation to *Clostridium perfringens* infection. Selenium, a non-metallic essential micronutrient, is able to modulate the immune response in chickens challenged first with *Eimeria maxima* (at 14 day posthatch) and then with *C. perfringens* (at 18 day posthatch). Treated groups received 10 and 20 μg of Selenium/egg and showed an increased serum antibody levels against *C. perfringens* α-toxin and NetB toxin and both lower intestinal lesion and oocyst production compared to the non-treated group, suggesting an amelioration of immunity response in the posthatch period [197]. In ovo injection at 12 days of incubation of a raffinose family oligosaccharides (RFO) extracted from *Lupinus luteus* seeds permitted a 2.5 log reduction in *C. perfringens* count and an 89% reduction in *Eimeria* spp. oocysts shedding [198].

In ovo administration of probiotics at day 18 of incubation permits a significant reduction in the severity of macroscopic lesions caused by *Eimeria* spp. in all intestinal segments and an improvement of the zootechnical performances [169]. Comparing the in ovo administration at day 12 of incubation, of a prebiotic composed by a trans-galactooligosaccharides (Bi^2^tos), to an antibiotic, given individually or together, it emerges that prebiotic, with or without the antibiotic supplementation, can reduce intestinal lesion and oocyst shedding in natural infected chickens [199]. After the in ovo injection of vitamin D_3_ and 25-hydroxyvitamin D_3_ (25OHD_3_) in chickens challenged at 14 days, it was observed that, in the group treated with the 25OHD_3_, compared to D3 and the control group, there was no reduction in performances that had previously been observed in the other groups [200].

## 8. Discussion and Conclusions

Nutraceuticals and phytonutrients have gained great interest in recent years, being presented as a possible alternative to antimicrobials and answering the high demand of antibiotic-free poultry products. A valid substitute for antibiotics must have similar properties in increasing growth performances, optimizing feed conversion and limiting infections from pathogens. Several natural compounds are currently available on the market as valid substitutes to antibiotics, and they are also able to stimulate the immune system, making the animals more resistant to infections.

In the last 10 years much research has been conducted on the use of nutraceuticals such as probiotics, prebiotics, vitamins, phytogenic extracts, and organic acids to verify their efficacy and safety against the most common GI pathogens of poultry. The results presented are sometimes conflicting; in fact, despite some very favourable effects, not all these natural substances have an efficacy in prevention or treatment of the disease. It is difficult to compare the various products, as dosages, methods of administration, age of the birds, and duration of administration are different in each trial. In any case, even when not able to directly inhibit the infection, the use of nutraceuticals and phytonutrients has proven to be of fundamental importance for GI health, minimizing pathogens’ effects. Most of the losses are due to subclinical form, and the enhancement in intestinal immunity is essential, allowing a reduction in morbidity and periods of non-weight gain of the animals. The reduction in the bacterial load and faecal elimination also represents an important element to reduce meat contamination during slaughter, and thus the spread of clinical forms in the final consumer. The in ovo technique is very promising, allowing the product to explain its effect before the chicken enters into contact with environmental pathogens, having a positive influence on influencing the development of GI and immune system, and thus leading to a further form of resistance to infections.

Regardless of the administration route, it is evident that, although not always able to be directly effective in the prevention of infections, nutraceuticals and phytonutrients allow a reduction in symptoms with an increase in the activity of the immune system and growth performances. Moreover, the diversity of the results presented in the various trials highlights the complexity of the effects deriving from the use of these compounds, and further studies will be necessary to highlight the mechanisms of action to make their use even more effective in poultry production.

## Figures and Tables

**Figure 1 animals-12-00892-f001:**
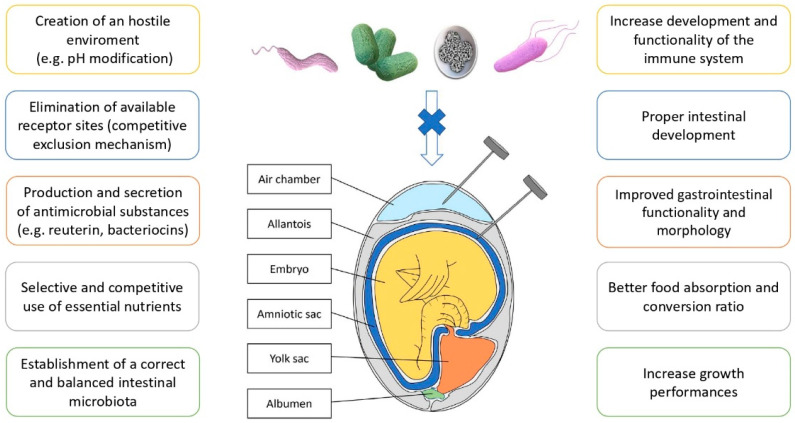
Benefits of in ovo inoculation of nutraceuticals and phytonutrient.

**Table 1 animals-12-00892-t001:** Classification of compounds that can be used in poultry production.

Compounds	Definition	Origin
Nutraceutical	A food (or its part) that provides medical or health benefits, including the prevention and/or treatment of a disease [17]	Plant or animal
Phytonutrient	Plant derived compound [18]	Plant
Phytochemical	A variety of plant-derived compounds with therapeutic activities such as anticarcinogenic, antimutagenic, anti-inflammatory, and antioxidant [19]	Plant
Bioactive Compound	Components in foods or dietary supplements, other than those necessary to the basic nutritional needs, which are responsible for changes in health status [20]	Plant or animal

**Table 2 animals-12-00892-t002:** Description of products for the regulation of the intestinal bacteria population in poultry and their principal effects.

Items	Definition	Mechanism of Action
Probiotics	Live microorganisms which, when administered in adequate amounts, confer a health benefit on the host [21]	Competitive exclusion Production of antimicrobial substances Stimulation of immune system Increased intestinal absorption surface Increased growth performance and feed intake Modulation of respiratory and GI microbiota [22,23,24,25,26,27,28,29,30,31]
Prebiotics	A nondigestible compound that, through its metabolization by microorganisms in the gut, modulates composition and/or activity of the gut microbiota, thus conferring a beneficial physiological effect on the host [32]	Nutrient source for the selective growth of beneficial bacteria of the intestinal microbiota Stimulation of short-chain fatty acids production Inhibition of bacterial adhesion to gut lining Change in mucin production Immunity boost Improvement in intestinal health and functionality. [15,33,34,35,36]
Vitamins	Vitamins are nutritional elements which are necessary for essential activities such as development, growth, and metabolism of cells [37]	Antioxidant effect Reduction in free radicals Increase in mucosal immunity Anti-inflammatory effect Immunostimulatory effects Increase in cellular immunity [37,38,39,40,41,42]
Phytogenic feed additives (or Phytobiotics)	Compounds of plant origin incorporated into animal feed to enhance livestock productivity through the improvement of digestibility, nutrient absorption, and elimination of intestinal pathogens [43]	Increase in growth performance, nutrient digestibility and gut health Introduction into the cell membrane of pathogens and consequent destruction with consequent ions leakage Antioxidant activity Modulation of intestinal microbiota composition [44,45,46,47,48,49]
Organic acids	Primarily composed of short-chain fatty acids (SCFA), also commonly referred to as volatile short-chain fatty acids (VSCFA), such as fumaric, propionic, acetic, lactic, butyric, and others. Other organic acids consist of medium-chain fatty acids (MCFA), and long-chain fatty acids (LCFA) [50]	Lowering pH of GI tract (reduction in acid sensitive bacteria) Potential for incorporation into cell membranes of target cells and promoting the loss of protons or cell ions (such as in Gram-positive bacteria) Promotion of gut health and performance [10,51,52,53,54]

## Data Availability

Not applicable.

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
