# Peer review of "The Role of Nutraceuticals and Phytonutrients in Chickens’ Gastrointestinal Diseases"

_animals, 2022, doi:10.3390/ani12070892_

Round 1

Reviewer 1 Report

In general, this morphological work is really interesting and well written. My minor comments aim to increase the scientific soundness and clarity of it.

  1. Title – The “gastro-enteric” is incorrect from anatomical point of view. Please use “gastro-intestinal” instead
  2. The topic of this article is not new and has been presented several times before. There are other articles dealing with the same subject (see below). To make this article more interesting to a reader I suggest the authors to precisely and clearly define the target question of this review.

See:

a) Shehata AA, Yalçın S, Latorre JD, Basiouni S, Attia YA, Abd El-Wahab A, Visscher C, El-Seedi HR, Huber C, Hafez HM, Eisenreich W, Tellez-Isaias G. Probiotics, Prebiotics, and Phytogenic Substances for Optimizing Gut Health in Poultry. Microorganisms. 2022 Feb 8;10(2):395. doi: 10.3390/microorganisms10020395. PMID: 35208851; PMCID: PMC8877156.

b) Swaggerty CL, Bortoluzzi C, Lee A, Eyng C, Pont GD, Kogut MH. Potential Replacements for Antibiotic Growth Promoters in Poultry: Interactions at the Gut Level and Their Impact on Host Immunity. Adv Exp Med Biol. 2022;1354:145-159. doi: 10.1007/978-3-030-85686-1_8. PMID: 34807441.

c) El-Shall NA, Abd El-Hack ME, Albaqami NM, Khafaga AF, Taha AE, Swelum AA, El-Saadony MT, Salem HM, El-Tahan AM, AbuQamar SF, El-Tarabily KA, Elbestawy AR. Phytochemical control of poultry coccidiosis: a review. Poult Sci. 2022 Jan;101(1):101542. doi: 10.1016/j.psj.2021.101542. Epub 2021 Oct 14. PMID: 34871985; PMCID: PMC8649401.

d) Umaya SR, Vijayalakshmi YC, Sejian V. Exploration of plant products and phytochemicals against aflatoxin toxicity in broiler chicken production: Present status. 2021 Sep;200:55-68. doi: 10.1016/j.toxicon.2021.06.017. Epub 2021 Jul 3. PMID: 34228958.

e) Madlala T, Okpeku M, Adeleke MA. Understanding the interactions between Eimeria infection and gut microbiota, towards the control of chicken coccidiosis: a review. 2021;28:48. doi: 10.1051/parasite/2021047. Epub 2021 Jun 2. PMID: 34076575; PMCID: PMC8171251.

  1. Lines 21, 28 and throughout the text – “gastrointestinal” should be acronymized to GI
  2. Line 129 – please consider to use “intraluminar pH of GI tract” instead “GI pH”. It is more accurate.
  3. Line 267 – “Medium Chain Fatty Acids” why this term is written in capital letters ? Moreover, this term already appear in line 161.
  4. Please unify the abbreviation codes. “medium chain fatty acids” are either acronymized to MCFAs (line 267, line 332), MCFA (table 2) or not acronymized (line 161).
  5. Line 362 – there is no such anatomical term as “intestinal tract”
  6. Line 527 – please change “gastro-enteric” into “GI”
  7. The term poultry is very wide and means birds that are bred for eggs and meats. But the authors actually refers to the chicken only. I would advice the authors to adapt the title.

Author Response

Rebuttal letter to Reviewer 1

In general, this morphological work is really interesting and well written. My minor comments aim to increase the scientific soundness and clarity of it.

Dear Reviewer, thanks for your kind comments.

  1. Title – The “gastro-enteric” is incorrect from anatomical point of view. Please use “gastro-intestinal” instead

We corrected the title.

  1. The topic of this article is not new and has been presented several times before. There are other articles dealing with the same subject (see below). To make this article more interesting to a reader I suggest the authors to precisely and clearly define the target question of this review.

We define the target question of this review at lines 66-68. Thanks for the articles, we added some of them to our review.

  1. Lines 21, 28 and throughout the text – “gastrointestinal” should be acronymized to GI Done.

  1. Line 129 – please consider to use “intraluminar pH of GI tract” instead “GI pH”. It is more accurate. Done, thanks!

  1. Line 267 – “Medium Chain Fatty Acids” why this term is written in capital letters?

           Moreover, this term already appear in line 161. Done

  1. Please unify the abbreviation codes. “medium chain fatty acids” are either acronymized to MCFAs (line 267, line 332), MCFA (table 2) or not acronymized (line 161). Done.

  1. Line 362 – there is no such anatomical term as “intestinal tract” Done.

  1. Line 527 – please change “gastro-enteric” into “GI” Done.

  1. The term poultry is very wide and means birds that are bred for eggs and meats. But the authors actually refers to the chicken only. I would advise the authors to adapt the title. Thank you for this suggestion, we adapted the title.

Reviewer 2 Report

This manuscript reviews the role of nutraceuticals & phytonutrients in poultry gastroenteric diseases.

The comments are as follows.

  1. Overall, the aim of the review is unclear. Did the review summarize the elimination effects of nutraceuticals or phytobiotics on intestinal pathogens? Or the mechanisms, the introduction of commercial products?
  2. The outcomes, such as growth promotion, FCR improvements, or recovery from the pathophysiological injury, because a major point of replacing alternative to AGPs is the points.
  3. More information on the general symptoms of chicken infected with Salmonella spp., C. perfringens, and Campylobacter jejuni are required.
  4. Table 1-2; I feel a strangeness for the definition of each item because it is unclear whether the items are for human or poultry (food or feed? Colonic microbiota for poultry?). The mechanism is also included in the definition. These tables need to be improved in terms of simpler and more brief definitions and the action mechanism for poultry.
  5. 16: Is this scientific announcement based on a reliable source? I don’t agree with the term “phytonutrient” because plant-derived compounds, such as resveratrol, is NOT a nutrient for human at least. We are alive without eating these compounds. The reference suggests such a term; however, it should avoid using it in this review.

Author Response

Rebuttal letter to Reviewer 2

This manuscript reviews the role of nutraceuticals & phytonutrients in poultry gastroenteric diseases.

Thanks for your useful comments.

 The comments are as follows.

  1. Overall, the aim of the review is unclear. Did the review summarize the elimination effects of nutraceuticals or phytobiotics on intestinal pathogens? Or the mechanisms, the introduction of commercial products?

The purpose of this review is to describe the different nutraceuticals or phytobiotics compounds (both commercial or not) that have been tested against the four pathogens described. Both the effects of elimination of the pathogen and the ability to reduce the consequences of the infection (e.g. reduction of the weight gain) were considered. In fact, even if beneficial effects deriving from the use of a specific product are observed, the underlying mechanisms of action are not always described.

In the introduction, line 66-68, the objective of the review is clarified.

  1. The outcomes, such as growth promotion, FCR improvements, or recovery from the pathophysiological injury, because a major point of replacing alternative to AGPs is the points.

The outcomes of this review shows that in most cases the alternative to AGP presented are able to display similar effect to it. In fact, even if always not directly able to reduce the infection, these compound can increase resistance to pathogens while also having growth-promoting effect (lines 92-95).

  1. More information on the general symptoms of chicken infected with Salmonella spp., C. perfringens, and Campylobacter jejuni are required.

We have included a more detailed description of the symptoms for each pathogen (lines 113-117; lines 224-226; lines 303-309).

  1. Table 1-2; I feel a strangeness for the definition of each item because it is unclear whether the items are for human or poultry (food or feed? Colonic microbiota for poultry?). The mechanism is also included in the definition. These tables need to be improved in terms of simpler and more brief definitions and the action mechanism for poultry.

We used the term food when refers to human or when included in definition based on human food. We corrected the term in feed, when used for poultry, in all the text.

We changed the definition of prebiotic and we tried to simplify the tables.                      

  1. 16: Is this scientific announcement based on a reliable source? I don’t agree with the term “phytonutrient” because plant-derived compounds, such as resveratrol, is NOT a nutrient for human at least. We are alive without eating these compounds. The reference suggests such a term; however, it should avoid using it in this review.  

Your opinion is indeed correct but this is a definition commonly used for Phytonutrient. We changed the previous references (16) citing the the definition used in another paper related to chickens (18).
